# Tracking Food Supply Chain Postharvest Losses on a Global Scale: The Development of the Postharvest Loss Information System

Thiago Guilherme Péra [1],*, Fernando Vinícius da Rocha [2] and José Vicente Caixeta Filho [1]

1 Luiz de Queiroz College of Agriculture (ESALQ/USP), University of São Paulo, Piracicaba 13418-900, Brazil; jose.caixeta@usp.br
2 Faculty of Animal Science and Food Engineering (FZEA/USP), University of São Paulo, Pirassununga 13635-900, Brazil; fernandorocha7@gmail.com
* Correspondence: thiago.pera@usp.br; Tel.: +55-19-3429-4580

**Abstract:** Reducing food losses presents an opportunity to enhance food security, minimize waste, and improve profitability within the production sector. Creating awareness among various stakeholders in the value chain about the significance of reducing postharvest losses is a fundamental step in this discussion. This article addresses the Postharvest Loss Information System (SIPPOC) development and applicability. SIPPOC encompasses tools designed to facilitate understanding food loss occurrences across different supply chain segments. The article provides insights into the tools incorporated within the information system and describes its historical background and protocol for database updates. In essence, SIPPOC enables the analysis of food loss throughout diverse logistical stages, thereby aiding in identifying critical points and implementing targeted actions for loss reduction. Drawing on SIPPOC data, the article further examines losses within the logistics chain by comparing potato, tomato, and mango agricultural productions.

**Keywords:** postharvest loss reduction; information system; database

## 1. Introduction

Multiple studies suggest that in order to meet the worldwide demand for food consumption, there will need to be a significant increase in global grain production in the upcoming decades [1–3]. However, an alternative solution would be to focus on decreasing losses throughout the entire value chain [1–4]. Reducing postharvest losses is crucial as it directly results in food and income loss for farmers and consumers globally. In addition to reducing food loss on the farm, it is important to implement policies that address the growing issue of food waste among consumers and retailers [4].

Reducing food loss and waste is challenging due to countries' infrastructure and technology disparities. Recent academic research is working on new technologies and management practices.

The academic literature on food loss and waste is based on quantifying it in various regions. Farmers' adoption of new practices, such as postharvest technologies, is influenced by cost, production scale, and access to rural extension services. Small-scale producers face challenges in adopting new technologies due to the high acquisition costs, lack of access to extension services, and the need to understand the potential benefits. Farmers also face significant difficulties due to limited access to markets and accurate information related to loss prevention.

A data-driven approach can be instrumental in preventing and mitigating postharvest loss, identifying critical points in the food chain, and developing more precise and effective solutions. From this perspective, information can be instrumental in formulating public policies and management practices to reduce losses in the food production chain, and it can be associated with the following:

1.  Data analysis, with the potential to identify patterns and trends that can help better understand the causes of losses and provide insights for policy development and management practices.
2.  Communication among food production and distribution stakeholders facilitates information sharing and collaboration to solve problems.
3.  Educational practices on loss reduction, where the data allow educators, policymakers, and industry leaders to pinpoint specific areas, offering a roadmap to develop targeted educational content and training programs.

This article introduces the Postharvest Loss Information System (SIPPOC) developed by the Research and Extension Group in Agroindustrial Logistics (ESALQ-LOG). In a world facing food scarcity and environmental concerns, SIPPOC innovates by providing insights into postharvest losses. The main objective of this paper is to present the tools available in SIPPOC, show details about how this information system was developed, and present an example to illustrate how SIPPOC can be helpful for future academic research.

This article is structured into three additional sections beyond this introduction. The second section briefly discusses the theoretical framework of this research line. Section 3 presents the key information regarding SIPPOC. Subsequently, the fourth section provides an example analysis using the information available in the SIPPOC database, comparing the losses of mango, tomato, and potato throughout the logistical stages. Finally, the Conclusion section summarizes the discussion presented, consolidating information on potential future directions.

## 2. Background

Studies addressing food losses in food chains have garnered attention in academia. With a growing number of published articles, the analysis objectives within this theme are diverse. The range of articles below illustrates the variety of types of analysis covering this topic.

For instance, Ref. [5] provides a better understanding of the differences between industrialized and developing countries and how these differences relate to postharvest losses. The author highlights that lack of infrastructure and low levels of technology are the main factors contributing to food loss during the harvest, postharvest, and processing stages in developing countries.

An essential portion of scientific articles studies packaging and other postharvest technologies. For example, Refs. [6–8] analyzed the effects of modified atmosphere packaging during refrigerated storage on food quality maintenance; Ref. [9] investigated ideal storage conditions for ripe mango fruits; and Ref. [10] examined the effect of clove oil on cassava starch films and the impact of these films on the postharvest shelf life of bananas. Postharvest treatments involving wax, among other types, are also effective in extending product shelf life [11,12].

Cold storage is the focus of [13]. The article demonstrates that this technology can play a crucial role in minimizing potato losses throughout the distribution chain.

These articles emphasize the importance of developing new packaging and postharvest techniques and how these advancements mitigate postharvest losses; Ref. [5] highlights that developing such technologies is one of the main factors in reducing food loss.

Also related to the development of new technologies, solar drying is an extensively researched topic in academia. The use of solar dryers during storage processes is the focus of [14]. The authors point out that the development of such tools can play a significant role for small-scale farmers, particularly in developing countries. Similarly, Ref. [15] analyzes dehydration techniques that could enhance nutritional quality and storage stability.

Parallel to this, the propensity of agents to adopt a new tool or technology can be a driving factor for postharvest reduction practices. Apart from technology availability, numerous other variables can hinder the adoption of new management practices, such as cultural factors, lack of information, and beliefs, among others.



In this context, Ref. [16] examined the relationship between the risk preferences of rice producers and the adoption of postharvest technology among farmers in Cambodia. The authors conclude that rice cultivation experience determines loss aversion, and more risk-averse farmers are likelier to use devices such as moisture meters.

Climate variables influence farmers' choices regarding improved storage technologies [17]. According to these authors, households that have experienced high rainfall in previous years are more likely to adopt preservation measures. Furthermore, farmers in climate-risk environments respond by implementing conservation measures against storage pests. Furthermore, Ref. [17] also emphasized the importance of access to extension services, as it increases the likelihood of adopting improved storage and preservation practices.

When examining the process of producing bananas, Ref. [18] found that certain factors contribute to postharvest losses (PHL) at the farm level. These include the leadership within the family, the size of the family, the portion of land dedicated to banana production, and the number of bananas produced each month. At the retail level, PHL is influenced by the gender of the seller and their association with groups. Therefore, the study emphasizes the importance of implementing comprehensive strategies that are sensitive to gender to reduce postharvest losses.

According to [4], various factors such as wealth, agricultural machinery, transportation, and telecommunications play a significant role in determining the amount of food lost after harvest in countries in the global south. The authors suggest that efforts to reduce food loss at the farm level in low-income countries should be combined with policies targeting the growing issue of food wastage among consumers and retailers.

The article by [19] supports that food waste in developing nations happens mainly during the supply chain due to insufficient technological infrastructure, resulting in significant postharvest losses.

Several studies in the academic literature have discussed the postharvest grain problem in African countries. Recognizing that reducing postharvest losses is a crucial strategy for ensuring food availability, Ref. [20] analyzed postharvest losses of maize in Africa. The author highlights the interconnection between research, extension services, agro-industry, marketing systems, and the political environment to address postharvest management issues. According to the author, inadequate postharvest management is one of the primary constraints to improving nutritional security in African countries.

Refs. [21–28] are examples of studies that have analyzed losses with applications to African locations. A general synthesis of these articles underscores the importance of an efficient storage system accessible to small-scale farmers, as improving access to postharvest tools and technologies is an effective way to mitigate food losses. These works converge to affirm that the producer can achieve more marketing opportunities, increasing profitability and the attractiveness of this economic activity. On the other hand, developing tools/technologies tailored to small-scale farmers poses a challenge related to production scale and financial viability.

Another grouping of works in this theme is related to the construction of mathematical models for monitoring and predicting the occurrence of losses. The objective is to determine the factors considered the main drivers of postharvest losses for different food and value chains.

For instance, Ref. [29] aimed to construct a grain loss prediction model by applying algorithms such as Support Vector Machine. With a similar approach, Ref. [30] identified India's main drivers of losses by analyzing the fruit and vegetable supply chain. These works list many possible critical factors for the occurrence of losses, such as lack of adequate storage facilities, inadequate handling of products on the farm and in markets, lack of proper packaging facilities, insufficient infrastructure, lack of processing facilities, lack of coordination between farmers and processing units, lack of forward integration from farmers to consumers, lack of linkages in the marketing channel from farmgate to the market due to small landholdings, a high number of intermediaries, and lack of knowledge

about the market demand. Ref. [31] presents recommendations for introducing innovative business models based on the results based on technology (use of information for better supply chain connectivity).

In addition to the exemplified themes above, another grouping of works is related to quantifying losses in logistic chains. These studies are connected to different production chains and countries, employing various methodologies. Works of this nature are the subject of analysis by the Postharvest Loss Information System (SIPPOC), discussed in the next section of this article.

## 3. SIPPOC: An information System for Food Losses

An information system is an organized set of elements that interact to collect, process, store, and disseminate data. It serves to support decision-making in various applications. As mentioned, this article section presents information about the Postharvest Loss Information System (SIPPOC).

SIPPOC is an information system dedicated to collecting, processing, and disseminating data related to postharvest losses in Brazil and worldwide. Its primary mission is to support public and private decision-making by providing a comprehensive database of postharvest loss information, integration with a geographic information system (GIS), loss calculation tools, and global loss indicator databases for various crops.

SIPPOC was developed by the Agroindustrial Logistics Research and Extension Group (ESALQ-LOG), initially with the support of the São Paulo Research Foundation (FAPESP) in 2017. The initial scope was to identify levels of postharvest losses based on primary information for the state of São Paulo, integrating this information with maps and logistics indicators, and developing a loss calculator tool for each type of logistics activity that quantified different spheres of losses: economic, energy, environmental, and nutritional. During this period, an initial survey consolidated a database on postharvest losses for various crops based on technical reports and scientific articles in the literature, focusing mainly on the state of São Paulo and Brazil.

With the support of the Consortium for Innovation in Postharvest Loss and Food Waste Reduction, the scope of SIPPOC was expanded to consolidate a global database of postharvest losses for different countries, crop types, and supply chain levels based on an extensive survey of technical and scientific articles on losses. This second phase of the Information System began in 2019, incorporating new data analysis tools into the platform.

### 3.1. Tools Available in SIPPOC

In its most recent version, SIPPOC consists of four tools. The first tool presented in this article is the Postharvest Loss Information Database. It is a database that contains information on postharvest losses in different countries, providing details on the supply chain links and the sources of information.

After the December 2022 update, the database comprises 5060 information on food losses, covering 88 countries and 22 agricultural production chains. The information is organized to allow for comparisons between different supply chain links, and all the data are derived from published academic works (a total of 277 articles)—see Figure 1.

Figure 2 displays the PHL database delineated by crop group, country, and individual crop alongside a map highlighting countries with the most processed data.

The objective of this tool is to enable users to make queries and analyses equipped with features that facilitate rapid data visualization and comparison. Figures 3 and 4 illustrate some of the possibilities incorporated into SIPPOC for data comparison.

Figure 3 showcases one of the features of SIPPOC, which enables the comparison of losses in different production chains across specific supply chain links. In this case, the circles represent the average value of losses based on the complete set of information included in the database. Figure 4 also compares the links within the logistics chain, highlighting the visualization of data spread (dispersion of information). This functionality

allows filters to select the specific crop of interest for analysis and determine how the data should be organized in plots (information per article or country).

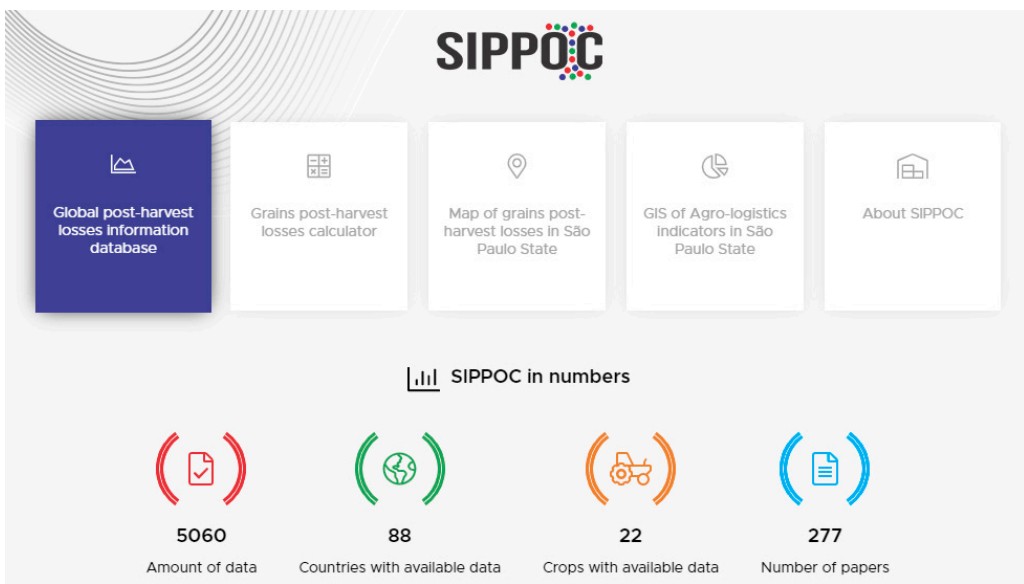

**Figure 1.** SIPPOC home screen.

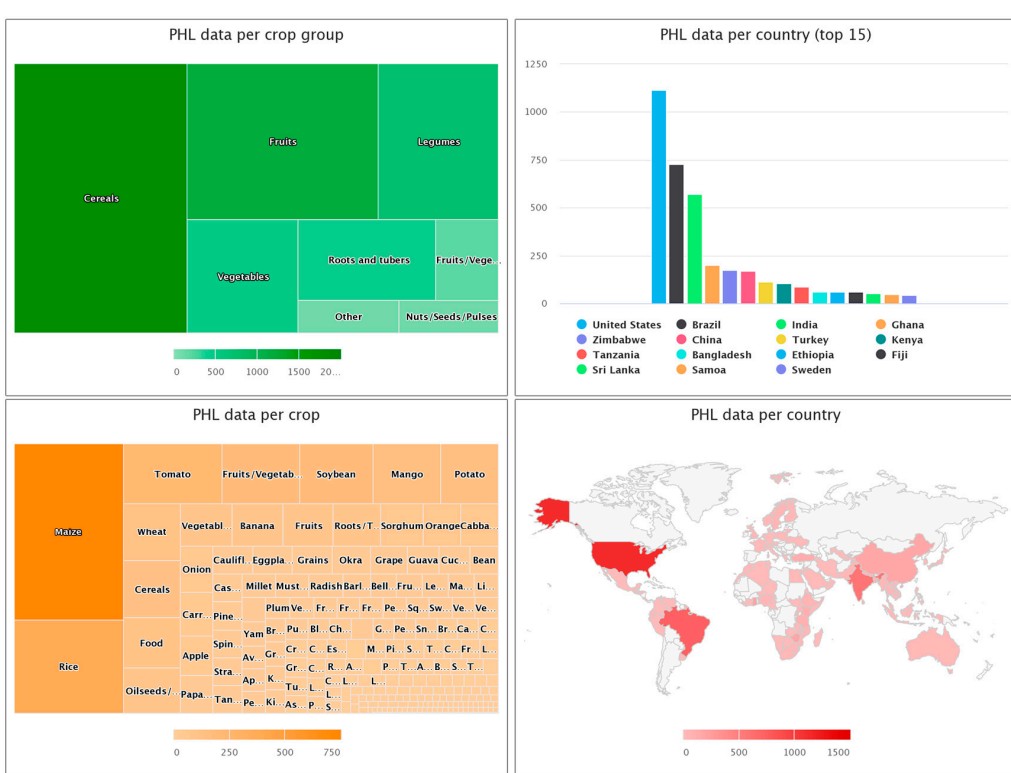

**Figure 2.** Indicators from the registered SIPPOC database.

Both functionalities allow users of the Information System to compare crop loss information. With these tools, it is possible to begin inferring the levels of existing losses and how divergent or convergent the studies in the field that quantify losses are. They also provide an initial understanding of which supply chain links are more susceptible to food losses.

**Average crop loss during different supply chain levels**

**Figure 3.** Comparison of food losses for different crops and logistics chain links based on the information incorporated in the SIPPOC database.

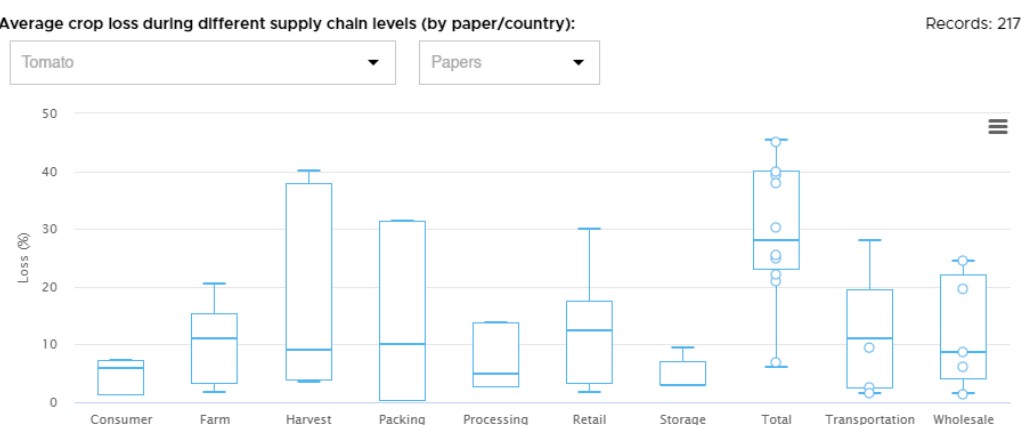

**Figure 4.** Comparison of tomato losses in different logistics chain links based on the information incorporated in the SIPPOC database.

Another functionality of SIPPOC is providing users access to the entire consolidated database on the platform. As shown in Figure 5, the database is available for download, containing detailed information about the articles (products, countries, supply chain links, loss quantification methods, publication year, average loss found by the study, and bibliography). Users can define filters (product, country, and supply chain link) in this feature, personalizing their query.

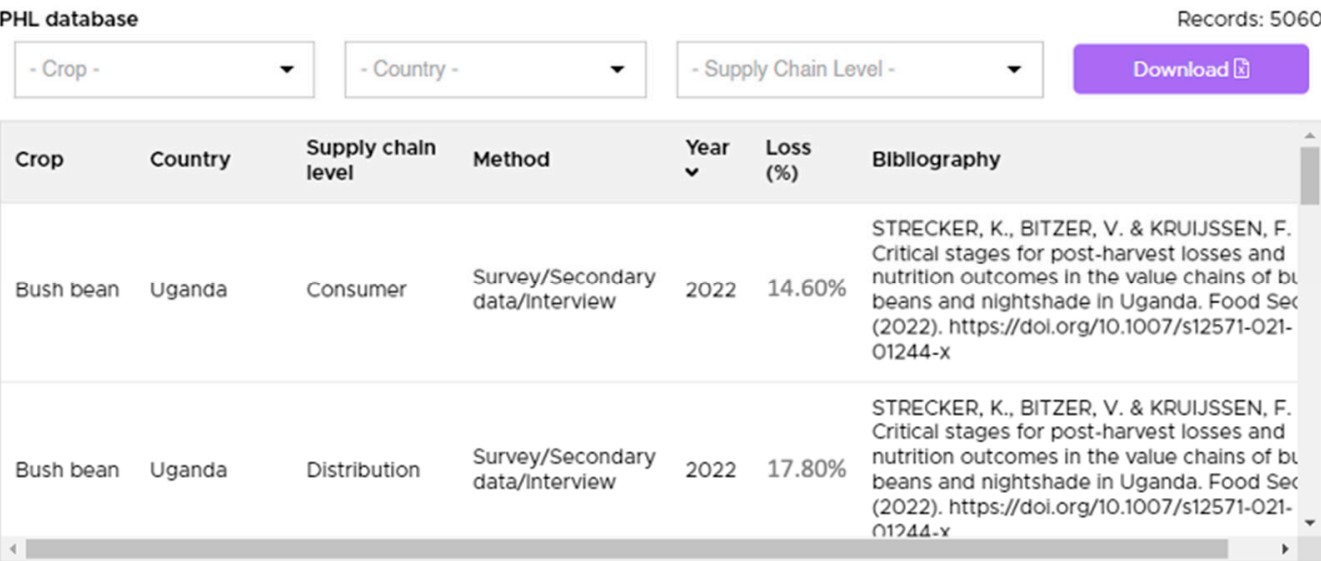

**Figure 5.** Database access screen, displaying information about the works included in SIPPOC.

Another tool is the Grain Logistics Loss Calculator. This tool allows users to calculate grain losses (soybeans, corn, and wheat) at different stages of Brazil's supply chain and analyze other losses beyond the physical ones, such as nutritional, economic, environmental, and energy-related (see Figure 6). The postharvest loss calculator input data is:

- Selection of the product available for loss diagnosis;
- Contemplated logistical activities: transportation between farm and warehouse, storage, transportation between farm and wholesale, among others;
- The average distance covered in road transport;
- Logistical costs involving transportation expenses;
- Product's market price.
- Based on these data, the generated outcomes include:
- Postharvest loss indicators by link and accumulated for the selected product;
- Economic loss indicator resulting from the opportunity cost of missed sales;
- Environmental loss indicator associated with the greenhouse gas emission level (carbon dioxide) expended in the logistics of the lost quantity;
- Nutritional loss indicator involving the quantity of nutrients that the product failed to provide;
- Land loss indicator intended for the production of lost products, calculated from the average observed productivity of each mesoregion of the analyzed product.

The tool suggests default loss percentages to the user based on information in the SIPPOC database. However, users can modify this information, customizing the analysis according to the loss standards they are interested in examining. The tool generates a series of numerical data and graphs as results, comparing levels of physical food losses and nutritional losses (carbohydrates, calories, and proteins).

The Map of Grain Losses in the State of São Paulo is a third tool incorporated into SIPPOC. It allows for spatial visualization through a map of food loss indicators specific to the State of São Paulo. The microregion of São Paulo State presents the loss information. Similarly, the Logistics Indicators Map for the State of São Paulo is another tool available in SIPPOC that enables spatial visualization of logistics-related indicators for the state (such as storage capacity, highways, etc.). Figure 7 portrays the "Map of Grain Losses in the State of São Paulo."

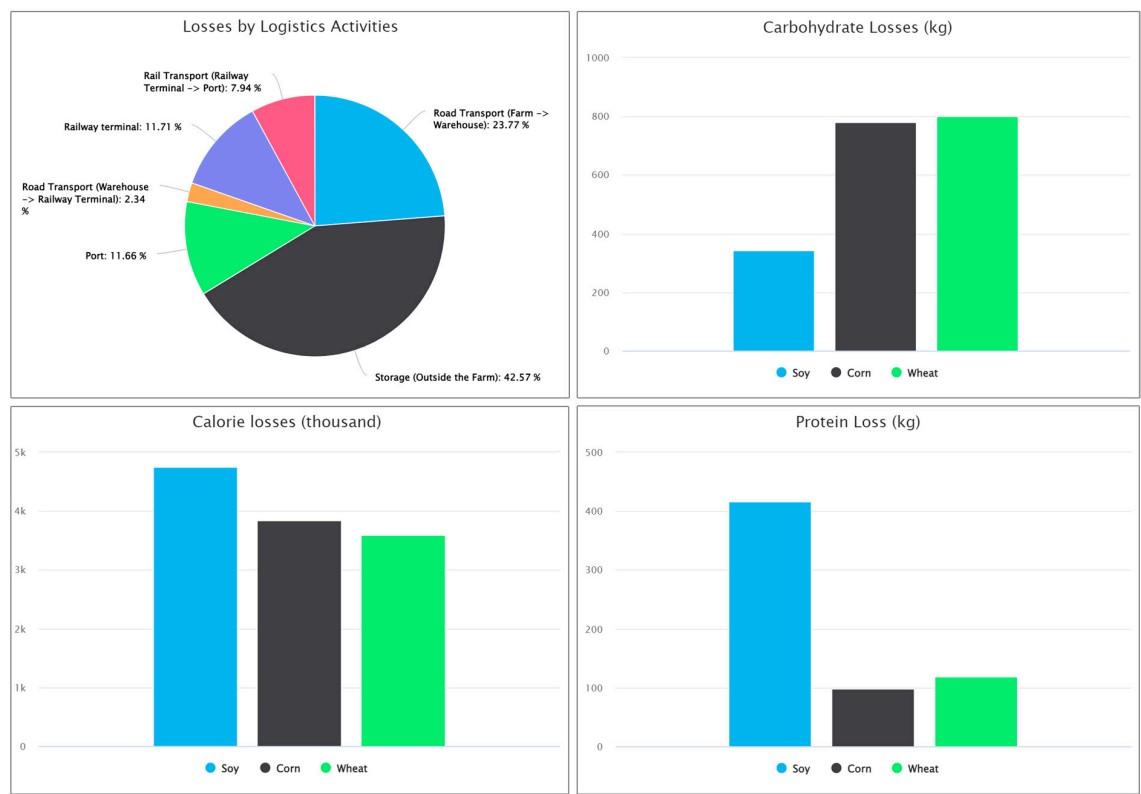

**Figure 6.** Grain Logistics Loss Calculator (Product: Grains, Transport operation: Multimodal, Destination: Export, Storage operation: outside the farm). Note: Due to approximations, the total may not add up to 100%.

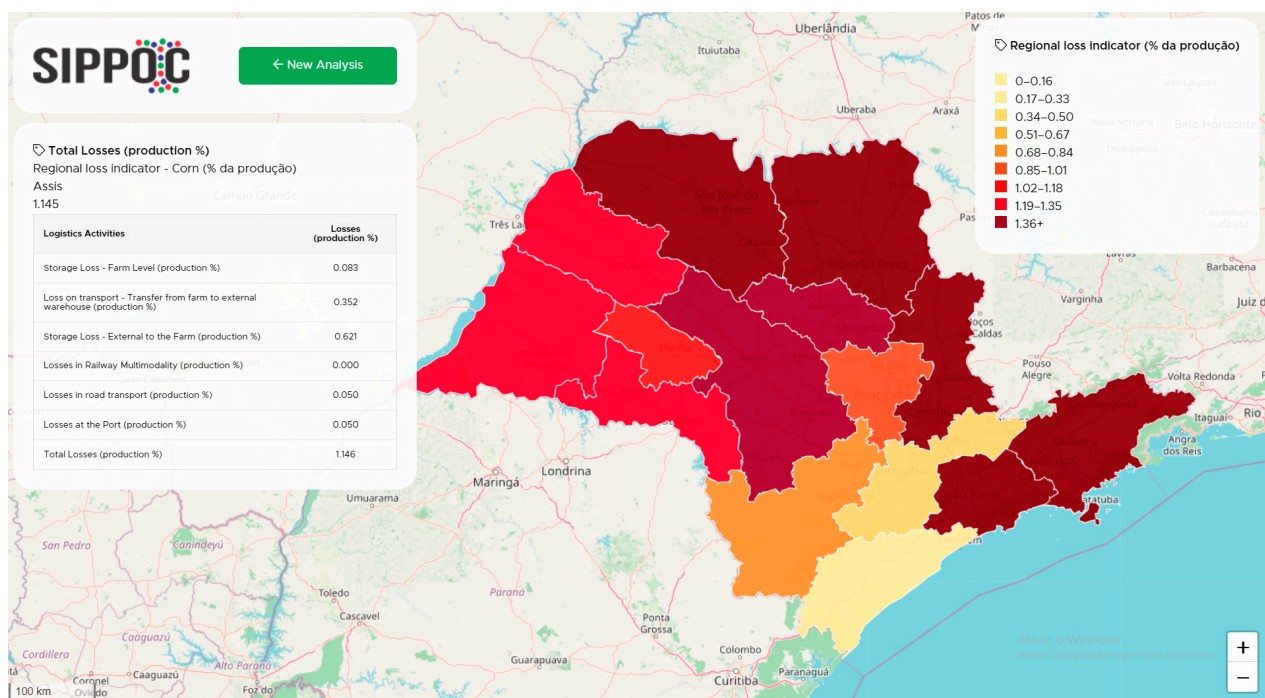

**Figure 7.** Example of the GIS incorporated into SIPPOC for visualizing loss and logistics indicators in the State of São Paulo.

### 3.2. Protocol for Information Updates

As mentioned earlier, the information on losses in the SIPPOC database originates from previously published academic papers. These papers are searched to obtain loss information through a systematic review protocol for postharvest loss information. Figure 8 summarizes the protocol for updating the postharvest loss indicator database of the platform.

**Figure 8.** Protocol for SIPPOC Database Information.

The starting point is the selection of databases for the article's research. As a standard practice, searches are performed in the following databases: Scopus, Web of Science, Science Direct, and Google Scholar.

Within these portals, the search for academic papers is carried out by combining keywords. The leading search terms are Postharvest Loss, Food Loss, Food Wastage, and Food Waste. These terms encompass the occurrence of food losses, allowing for information retrieval at various stages of the logistics chain.

Many papers are found on various platforms when searching for information on the quantification of food losses. However, not all of these papers are relevant and useful. To determine if a paper is relevant, we analyze the title and abstract to identify the scope of the study. One main criterion for a paper to be included in the SIPPOC information database is that it quantifies the postharvest loss of one or more products throughout the supply chain. Additionally, the database aims to provide details on the methods used to quantify losses in each article, such as sampling and direct weighing.

Once the articles have been selected based on the criteria above, they are further analyzed to extract relevant information for inclusion in the information system. This process involves obtaining information on the analyzed crop (agricultural production), country, logistical stage, range of quantified losses, method of quantifying losses, and other bibliographic details.

The SIPPOC database update occurs semiannually based on this systematic approach. Additionally, direct submissions of papers by academics interested in making their articles available for consultation through SIPPOC are also permitted.

## 4. Analysis of Losses in the Supply Chains of Potato, Mango, and Tomato

As designed, SIPPOC serves as a tool allied to the analysis of postharvest loss information for a significant set of crops across different countries. As previously summarized, this data-driven approach can assist in identifying patterns and trends for a better understanding of the subject while potentially supporting various research and educational practices.

This section compares the observed differences in losses for three essential crops in the Brazilian context: potato, mango, and tomato. The data came from the SIPPOC database.

Table 1 presents information on food losses at different logistic stages, including harvesting, storage, transportation, wholesale/retail, and consumer levels, for the three products: potato, mango, and tomato. Figure 9 illustrates this information, allowing inferences regarding the variability of the data.

**Table 1.** Potato, mango, and tomato losses at harvesting, storage, transportation, wholesale/retail, and consumer levels.

| Product | Logistic Stage | First Quartile | Median | Mean | Third Quartile |
|---------|----------------|----------------|--------|------|----------------|
| Potato | Harvest | 1.9% | 5.7% | 7.9% | 8.2% |
| | Storage | 8.3% | 14.6% | 22.3% | 32.0% |
| | Transportation | 5.3% | 9.2% | 9.2% | 13.1% |
| | Wholesale/Retail | 2.0% | 6.5% | 5.9% | 8.3% |
| | Consumer | 2.5% | 6.4% | 7.8% | 13.0% |
| Mango | Harvest | 2.9% | 4.4% | 14.4% | 8.8% |
| | Storage | 4.8% | 5.3% | 5.3% | 5.6% |
| | Transportation | 8.0% | 10.6% | 12.4% | 18.2% |
| | Wholesale/Retail | 5.3% | 10.5% | 13.0% | 18.1% |
| | Consumer | 5.6% | 11.5% | 12.0% | 17.1% |
| Tomato | Harvest | 4.9% | 8.9% | 17.8% | 27.8% |
| | Storage | 3.0% | 3.3% | 4.8% | 6.0% |
| | Transportation | 2.2% | 5.0% | 10.3% | 14.4% |
| | Wholesale/Retail | 3.4% | 12.8% | 12.9% | 15.6% |
| | Consumer | 1.3% | 5.0% | 4.3% | 7.0% |

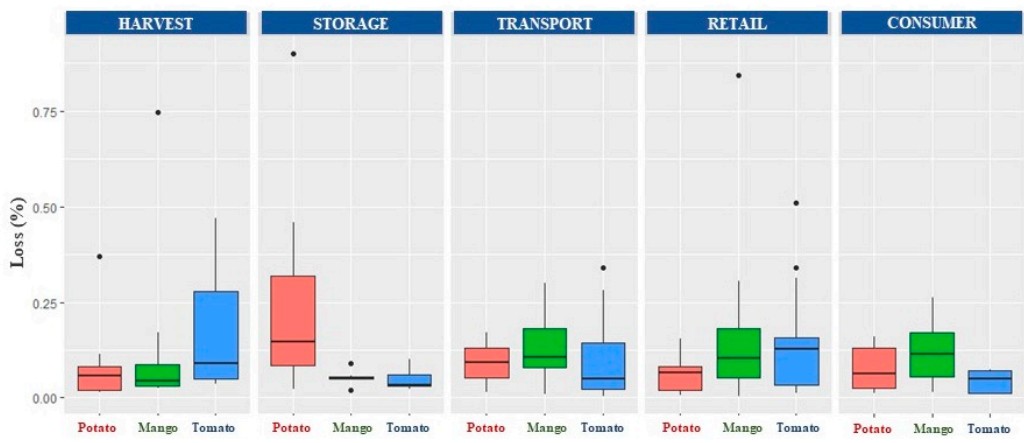

**Figure 9.** Losses of potato, mango, and tomato at harvest, storage, transportation, wholesale/retail, and consumer.

Based on the presented information, in the harvest stage, potatoes had the lowest average losses (7.9%), followed by mango (14.4%) and tomato (17.8%). The median of potato (5.7%) and mango (4.4%) was lower than that of tomato (8.9%), indicating that half of the potato and mango data fell below these values. The data variation (difference between quartile 3 and quartile 1) was lower for mango (5.9%), followed by potato (6.3%) and tomato (22.9%). Among the products, tomatoes showed the highest variation and median, indicating that losses in the harvest stage of this product can be more significant in some cases.

In the storage stage, mango had the lowest average losses (5.2%), followed by tomato (4.8%) and potato (22.3%). The median of tomatoes (3.3%) was lower than that of potatoes (14.6%) and mango (5.3%). The data variation was higher for potatoes (23.7%), followed by tomatoes (3.0%) and mango (0.8%). Therefore, in the storage stage, mangoes performed

better regarding average and data variation, while potatoes had the highest deviation and median.

In the transportation stage, the median of potatoes (9.2%) and tomatoes (5.0%) was lower than that of mangoes (10.6%). The data variation was more down for potatoes (7.8%). This indicates that the potato transportation loss data were more concentrated around the median, while mangoes and tomatoes showed more dispersion.

In the wholesale/retail stage, mangoes had the highest average losses (13.0%). The middle for potatoes was 5.9%, and for tomatoes it was 12.9%. The median of potatoes (6.5%) was lower than that of mangoes (10.5%) and tomatoes (12.8%). In contrast to potatoes, the loss data for mangoes and tomatoes in this logistic stage were more dispersed than the median.

In the consumer stage, potatoes had an average loss of 7.8%, mangoes had 12.0%, and tomatoes had 4.3%. The data indicate that mango losses in the consumer stage were more dispersed than the median (17.1% difference between the third and first quartile). Tomatoes performed better in terms of data variation and median.

Considering the results of the analyses, with a specific focus on the potato chain, the storage stage had the highest average losses (22.3%) and the highest data variation (23.7%). This finding indicates that the storage stage is the weakest link in the potato logistics chain. Reducing losses during storage requires investments in appropriate infrastructure, such as refrigeration systems and humidity control, as well as constant monitoring of storage conditions. It is also essential to adopt good handling and storage practices to prevent physical damage to potatoes.

For the mango supply chain, the harvest, wholesale/retail, and transportation stages had the highest average values for losses—14.4%, 13.0%, and 12.4%, respectively. Among these, the transportation and commercialization phases deserve greater attention due to the higher variability of the information. Specific actions could include investment in appropriate packaging to protect mangoes from physical damage and temperature variations. Additionally, monitoring transportation conditions and adopting more efficient routes to reduce travel time is essential.

For tomatoes, the harvest stage had the highest average losses (17.8%) and the highest data variation (22.9%), indicating that this stage is the weakest link in the logistics chain of this agricultural crop. Reducing losses in this stage involves adopting good farming practices, such as using more resistant varieties, employing appropriate management techniques, and training workers engaged in the harvest.

These findings underscore the need for a holistic approach to address postharvest losses. With a blend of technology, training, and infrastructure investment, it is possible to minimize these inefficiencies. Such efforts result in economic benefits and contribute to sustainability and food security objectives.

Adopting monitoring and tracking technologies for all stages of logistics and analyzed products can help identify critical points in the logistics chain and enable prompt intervention in case of issues. Moreover, it is essential to involve all actors in the logistics chain, from producers to end consumers, to raise awareness about the importance of reducing losses and adopting good management and consumption practices.

Different crops have varying needs and challenges, making a one-size-fits-all solution impossible. Factors such as perishability and logistical challenges differ across regions. Strategies that work for one crop may not work for another due to their unique postharvest requirements and vulnerabilities. A context-specific approach, tailored to each food and production locale, is necessary.

## 5. Conclusions

Studies on food losses in food chains have been growing in academia, addressing various analyses and objectives, such as understanding the differences between developed and developing countries, postharvest technologies and packaging, and adopting new management practices. Focused on quantification, studies analyzing postharvest losses

in food chains are essential for identifying critical points and serve as a starting point for developing a series of other research works and policies to mitigate losses.

The Postharvest Loss Information System (SIPPOC) is a tool to improve the management of agricultural supply chains by aggregating a broad set of information on the subject and facilitating the analysis of loss occurrence at different logistical stages. The design and implementation of this system focused on developing tools that could contribute to various studies on this subject, serving as a starting point for identifying the leading causes of losses and guiding improvement actions to reduce waste and increase sector efficiency. SIPPOC is a tool that can provide information support to policymakers, managers, and researchers on postharvest losses, contributing to the sustainable development goals of the United Nations, precisely goal 12 (Responsible Consumption and Production). As SIPPOC is a repository of loss data for various products, supply chain levels, and countries, a range of analyses can be performed using the existing tools on the platform and by cross-referencing the platform's information with other databases. For future work, we recommend cross-referencing SIPPOC information with other databases to conduct analyses such as determining factors of postharvest losses for products and countries of interest, efficiency analysis of loss management in different countries, cluster analysis of losses, and common factors associated with countries, among others.

SIPPOC fills an essential gap by consolidating postharvest loss information to understand losses and enable comparisons between countries, crops, and supply chain levels. Additionally, the tool provides users with organized access to the collection of technical-scientific publications that make up the database. It is understood that such a tool can be an essential contribution to fostering research on public policies and food security.

As we previously discussed, reducing losses in agriculture and postharvest processes is closely linked to using data-driven educational methods. By analyzing data to identify where and how losses occur, we can develop targeted training modules that equip individuals in the food production industry with the knowledge and skills needed to minimize waste. This approach not only helps to establish effective management practices and public policies, but it also serves as a vital tool for creating and improving educational content.

While SIPPOC offers valuable data and analysis capabilities, it is crucial to recognize that the accuracy of its information relies heavily on the quality and availability of academic papers and research data. Furthermore, SIPPOC relies primarily on published academic works, potentially overlooking unpublished research or real-time data. Recognizing these limitations is essential for users to make informed decisions and to drive continuous improvement in addressing food loss challenges.

**Author Contributions:** Conceptualization, T.G.P. and F.V.d.R.; Methodology, T.G.P. and F.V.d.R.; Formal analysis, F.V.d.R.; Investigation, F.V.d.R.; Data curation, T.G.P. and F.V.d.R.; Writing—original draft, T.G.P. and F.V.d.R.; Writing—review & editing, F.V.d.R.; Supervision, J.V.C.F.; Project administration, J.V.C.F.; Funding acquisition, T.G.P. and J.V.C.F.. All authors have read and agreed to the published version of the manuscript.

**Funding:** São Paulo Research Foundation (FAPESP)—(grant number 2015/22097-1), Consortium for Innovation in Postharvest Loss andFood Waste partially funded the development of SIPPOC. and Luiz de Queiroz Agrarian Studies Foundation (FEALQ) for financial assistance in publishing the article.

**Institutional Review Board Statement:** Not applicable.

**Data Availability Statement:** SIPPOC data can be consulted at: https://sippoc.esalqlog.com.br/en/login, accessed on 17 August 2023.

**Acknowledgments:** The authors acknowledge the researchers of the Group of Research and Extension in Agroindustrial Logistics (ESALQ-LOG) that contributed to the development of SIPPOC.

**Conflicts of Interest:** The authors declare no conflict of interest.

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
