# Peer review of "Tracking Food Supply Chain Postharvest Losses on a Global Scale: The Development of the Postharvest Loss Information System"

_agriculture, doi:10.3390/agriculture13101990_

Round 1

Reviewer 1 Report

Introduction:

The introduction effectively sets the context for the paper. However, it would be beneficial to include a concise statement of the paper's objectives and research questions to guide readers.

 A brief overview or roadmap of the paper's objectives and content in the introduction would provide readers with a clearer sense of what to expect.

Background:

The background section provides an extensive review of related studies on food loss. While informative, it could benefit from more concise summarization of the key findings and contributions of each referenced study.

Ensure that each reference is properly cited and included in the reference list.

SIPPOC: An information system for food losses:

This section offers a detailed description of SIPPOC and its tools. To improve accessibility, consider providing a brief overview of SIPPOC's objectives and its contribution to food loss reduction before delving into its tools and protocols.

Analysis of losses in the supply chains:

The analysis section is valuable for highlighting the differences in losses for different products and supply chain stages. It effectively utilizes tables and figures to present the data. To enhance the discussion, consider providing interpretations or implications of these findings.

Conclusion:

The conclusion should succinctly summarize the paper's key findings and offer insights into potential future directions more explicitly.

The paper is generally well-written, but some sentences are lengthy and complex. Breaking them into smaller, more comprehensible sentences would enhance clarity. Ensure consistent terminology and formatting throughout the paper.

Author Response

Thank you for reviewing our article.

We have incorporated the suggested changes in the introduction and literature review.

We have included more details regarding SIPPOC.

We have expanded the discussion on the comparative analysis of agricultural crops in the final section.

We have conducted a thorough English language review.

Reviewer 2 Report

I congratulate you for addressing a very necessary issue, since the extent of food losses in the food chain is still very unknown. Your effort and proposal are positively appreciated.

Even so, there are aspects that can clearly be improved. I try to indicate some ideas for you to contemplate:

In the part of the introduction (lines 23 to 33) general arguments are given, some would have to be supported with bibliographical citations.

When talking about the terms "losses" and "food losses and waste" (line 34) it is not clearly identified if they are talking about the same reality or different issues. As you know, definitions are especially sensitive and can change the meaning of the text.

In the points that they mention to formulate public policies, aspects 2 and 3 seem the same, although they have been separated. For its part, aspect 4 on educational practices, I think that could be specified a little more, what it refers to in this case.

In the Background point, many times when reading this section, it feels more like a listing of papers rather than a true collective analysis. There is a need for a more in-depth examination of the documents and the creation of a more solid argumentative thread.

Figure 3,  I'm not sure if it would be possible to create a chart that shows a sequence between the stages depicted in the image. For instance, having "total" at the end of the chart, retail next to wholesale, or specifying whether the word "Farm" includes the harvesting phase or not, etc. Thinking in a more logical order like the one that follows the chain, I think it would help to make the figure more comprehensive.

Figure 4, this part of the tool is interesting, but one issue that could help the comparability of results is the inclusion in the tool of the concept of food losses and/or waste, and that it reflects how it has been used in each study.

Please, explain a little better about the example of Figure 6, to know on the map, what is meant by "regional loss indicator".

Line 232, personally for new searches that can be done in the future, I would also use the term "Food Wastage".

To your comment that begins on line 245, I personally miss some type of analysis in relation to the number of papers already included in the tool.

Table 1, I would include the number of papers or "n" used to calculate the mean and median. I would also include maximum and minimum values, as well as, if possible, some additional data dispersion analysis or error margin estimation.

Next to Table 1, I think it is not necessary to describe all the results of the table, since for that we have the table, which is also reinforced with Figure 7. Starting from line 272, I would describe the most striking results, parameters that they can be deduced and/or calculated from the table etc., making it a richer discussion than just stating the results shown above.

Along with the final conclusions, or before them, in a study like this where a new tool is presented and which aspires to provide important solutions to the problem of waste, it is usually advisable to make clear the limitations of the work, or of the proposed tool.

Author Response

Thank you for your valuable feedback, which has greatly improved the quality of our manuscript.

For the comments, we:

  1. Added references in the introduction chapter.
  2. Expanded the discussion on the relationship between SIPPOC development and its potential use in educational practices regarding post-harvest losses.
  3. Provided further exploration of the academic works mentioned in the theoretical framework chapter.
  4. Regarding Figure 3, we have opted to keep it as is since it is an image directly extracted from SIPPOC. However, we have accepted this suggestion for improvements in SIPPOC in the next update cycle. Thank you on behalf of the ESALQ-LOG Group. We also appreciate the relevant suggestion for Figure 4.
  5. Clarified the meaning of "Regional Loss Indicator" (losses are calculated by micro-region in the State of São Paulo).
  6. We also appreciate the inclusion of the term "Food Wastage" as part of the search terms in the upcoming update cycles. We will update the protocol accordingly.
  7. Made other adjustments in discussing the results in the comparative analysis between crops.
  8. Added sections addressing the limitations of the information system

Reviewer 3 Report

This paper addresses the applicability of the Postharvest Loss Information System and describes its background and protocol for database updates. However, the following issues need to be further discussed.

1. The logical structure of the introduction is not well organized. The reader cannot clearly understand the necessity and innovation of this paper.

2. The literature is too old, and the author should pay attention to recent developments in existing research and discuss existing literature in more depth.

3. Figure 6 shows the food losses in different regions, is this an average annual loss? And logistics indicators don't seem to be represented in the graph.

4. In section 3.2, "The focus is on studies that analyze loss... and other bibliographic details.", the description is too redundant, it is recommended to re-describe.

5. Food losses will be different under different storage and transportation conditions, does the author take into account food losses under different conditions?

6. The interpretation of the numerical results is simple and insufficient, and I suggest that the author enrich the interpretation of the results.

Minor editing of English language required

Author Response

Thank you for reviewing the paper and providing feedback. Regarding each of the points:

  1. We have revised the logical structure of the introduction and other parts of the article.

  2. We have added several additional works to the literature review, as well as delved into more details about some of the previously mentioned works. In any case, the goal here was to analyze key works among the wide range of topics related to post-harvest losses.

  3. We have adjusted the text to make this point clearer.

  4. We have rewritten part of this section to improve clarity.

  5. Yes, we are aware that losses vary significantly due to different storage and transportation conditions, for example. However, SIPPOC is a tool designed to aggregate this information without necessarily making such differentiations.

  6. We have expanded this discussion as well.

Additionally, we want to mention that we have conducted textual revisions throughout the entire article.

Reviewer 4 Report

Dear Authors.,

Despite the effort to introduce the SIPPOC framework as a means to generate valuable insights for food loss management, this article deviates from the typical objectives of a scientific or research endeavor. It does not appear to have a defined research goal aimed at exploring specific questions or testing hypotheses. Furthermore, it does not provide actionable information for a global community approach to administration and lacks the identification of barriers and conditions necessary for its global applicability.

1.       The introduction of a scientific paper should ideally conclude with a summary by the authors, ensuring that the issue is introduced and explained clearly. However, this essential summarization and clarification of the problem is noticeably absent in this section.

2.       The background information presented appears to lack cohesion and a clear theoretical framework. It seems to consist of a collection of literature reviews without effectively connecting them to one another. Furthermore, the diverse conditions and supply chains associated with different agricultural products, such as mangoes and rice, make it challenging to draw direct comparisons between the postharvest losses of these distinct commodities. As a result, the author has not succeeded in presenting a coherent and consistent argument in this section.

3.       The paper's objective is unclear. It remains uncertain whether the authors intend to present information generated within an established framework or if they seek to create a novel information system aimed at aiding decision-making in the context of food waste management.

4.     The paper lacks adherence to a clearly defined scientific methodology that would provide substantial support for the obtained results and the eventual conclusions drawn.

5.       The primary emphasis of the paper lies in the introduction of a database called SIPPOC, designed for monitoring various research efforts related to food loss globally. A significant challenge arises from the fact that numerous papers and reports are composed in languages other than English. The question at hand pertains to how this database can effectively obtain and analyze information from these non-English sources.

6.     What are the benefits of including the outcomes related to food loss for a limited selection of crops within this article?

The Language needs minor revision.

Author Response

Dear reviewer,

Thank you for the reading and suggestions for improving the article. We have revised the document to address the points mentioned, specifically:

  • We revised the abstract and made some changes for better alignment with the theoretical framework
  • We clarified the article's objectives more prominently
  • And we highlighted more clearly the gaps that SIPPOC intends to fill.

Thank you.

Round 2

Reviewer 2 Report

I appreciate the improvements made to the text and the notes you have taken for future tool enhancements. I would like to add a few remarks, in case you can improve them:

1)      Considering the significance of bibliographic citations for the proper functioning of your tool, I am surprised that you have not included any in the introduction.

2)      Regarding my comment on the results detailed in Table 1, specifically the loss figures for tomato, mango, and potato, I find it somewhat simplistic. It would have been beneficial to enrich this example by explaining what results we can expect from SIPPOC. This could include insights into the challenges faced when establishing Figure 7 (now 9), the dispersion of primary sources of information, or the territorial scope of the analyzed works. In addition, commenting on more than just the results in percentages would be helpful.

3)      Perhaps they can make global improvements to better showcase the objectives, approaches, and purposes of the SIPPOC tool and how much it can contribute to the issue of crop losses and food waste. Keep in mind that readers who are not accustomed to these topics may have gaps in understanding its scope. If they make this effort, their reading will be more engaging, as it truly holds more value than may be perceived from the article's text.

My perception is that your tool is good and will likely become even more efficient as the supporting database grows richer and more rigorous. You are facing an extraordinary challenge, particularly in terms of maintaining coherent data comparisons to ensure it remains up to date. I send you all my encouragement and best wishes for this significant undertaking.

Author Response

Dear reviewer,

Thank you once again for the comments on the paper.

We have adjusted better in the introduction chapter, adding some bibliographic references in this section. We have also added pertinent comments to Figure 9 and tried to make the types of contributions SIPPOC can offer more evident.

Thank you.

Reviewer 4 Report

Dear authors,

You have taken into consideration all the comments provided and integrated them into the manuscript. However, I would like to suggest the addition of a section after the conclusion, where you can openly address some of the remaining limitations of the study and provide a concise explanation of these limitations.

It is acceptable

Author Response

Dear reviewer,

Thank you once again for the comments regarding our paper.

Once more, your contributions have been very valuable for the improvement of the work. Regarding a specific section for the limitations of the study, we still believe it is best to place this information at the end of the conclusions section. Upon revisiting the text, we believe that a specific section would be out of place within the overall structure of the paper.

Thank you again.